# Victim Detection and Localization in Emergencies

**DOI:** 10.3390/s22218433

**Published:** 2022-11-02

**Authors:** Carlos S. Álvarez-Merino, Emil J. Khatib, Hao Qiang Luo-Chen, Raquel Barco

**Affiliations:** University Institute for Telecommunication Research (TELMA), CEI Andalucia TECH E.T.S.I. Telecommunication, University of Malaga, Bulevar Louis Pasteur 35, 29071 Malaga, Spain

**Keywords:** detection, localization, UWB, WiFi FTM, victims

## Abstract

Detecting and locating victims in emergency scenarios comprise one of the most powerful tools to save lives. Fast actions are crucial for victims because time is running against them. Radio devices are currently omnipresent within the physical proximity of most people and allow locating buried victims in catastrophic scenarios. In this work, we present the benefits of using WiFi Fine Time Measurement (FTM), Ultra-Wide Band (UWB), and fusion technologies to locate victims under rubble. Integrating WiFi FTM and UWB in a drone may cover vast areas in a short time. Moreover, the detection capacity of WiFi and UWB for finding individuals is also compared. These findings are then used to propose a method for detecting and locating victims in disaster scenarios.

## 1. Introduction

Disaster management is a topic of the utmost importance in modern society. Naturally, as Information and Communication Technologies (ICTs) progress, novel applications are found for disaster management. These applications are subject to very challenging environments in disasters, where existing infrastructure (such as Base Stations (BS) or Access Points (APs)) is often inaccessible, time is a limited resource, and danger for rescuers and victims is present at all times.

One of the most challenging tasks in a disaster scenario is the detection and localization of victims, especially in disasters that involve being trapped under rubble. Detection and localization comprise the first in a series of steps [1], which also include the assessment of the victim’s status, communication, release and, transfer to a safe localization. The localization task is performed traditionally either with direct observation by the first responders (often with prior approximate information on where the victims might be located) [2] or with trained canine units [3].

With the emergence of wireless networks, many new possible applications for disaster management are enabled. Firstly, mobile networks provide an infrastructure-free connectivity from the point of view of the users, making the deployment of connected first responder assistance equipment immediate and simple. Secondly, since connected devices are ubiquitous, the victims are also often within reach of (or very close to) them, increasing the chances of detecting, locating, or communicating with them. To successfully localize victims within a disaster scenario, a gross estimation with a precision below 200 m is required [4]. This requirement is more relaxed than localization for other common location-based services currently under research (such as self-driving vehicles [5] or augmented reality [6], which demand centimeter-level accuracy); however, the context of the devices is much more challenging (e.g., with partially operational infrastructure or under rubble).

In the next few years, most of the smartphones will integrate UWB chipsets [7,8] and/or support the WiFi FTM protocol [9]. Following the trend of the market and knowing the advantages that both technologies bring for indoor positioning, UWB and WiFi FTM are likely to become the de facto positioning technologies for indoors [10,11,12,13]. From the victims’ perspective, the tendency of the integration of UWB and WiFi FTM in the chipsets makes the cost associated with this integration virtually null.

The contributions of this paper are the study of the role of wireless networks, in particular UWB and WiFi FTM, in the detection and localization of victims in disaster scenarios, specifically under rubble, providing an overview of the existing methods, their degree of implementation, and their precision. In addition, in this work, a scheme for opportunistically locating devices using several technologies for application in emergencies and a proof-of-concept using real devices are presented.

The rest of the paper is organized as follows. In Section 2, an overview of the challenges in locating victims in disasters is provided along with a review of existing solutions. Section 3 provides an overview of the different location technologies, explaining their features used in this work. In Section 5, the proposed opportunistic fusion method is described, along with the particularities of using it in emergency scenarios. The proof-of-concept and the scenario are described in Section 6. In Section 7, the results are presented and discussed. Finally, the conclusions are reviewed in Section 9.

## 2. Challenges of Detection and Localization of Victims under Rubble

Rescuing people in the first 48 h, also known as the golden hours [14], is crucial. After this time period, the chances of survival drop off drastically. Thus, a fast and efficient deployment is essential in these emergency cases. In this section, an overview of the challenges of locating victims and the methods used in the real world is provided.

There are many different kinds of disasters that imply having to find and rescue people in hazardous environments. Some typical examples are earthquakes, mass or individual transportation accidents, terrorist attacks, or extreme weather. When one of these situations occurs, the environment where the victims are located suffers rapid and traumatic changes, such as building collapses, flash floods, or fires. Of course, no two disasters are alike, since the victim’s status, the resources available for the first responders, and the dangers for both depend on very scenario-specific factors, sometimes even depending on the personal traits of the involved actors. Consequently, victims may be located in many different places, depending on the type of disaster and the specific factors.

Finding victims in these situations is especially challenging, and while the challenges are very situation-specific, some general situations can be described:Victims are in hard-to-reach places, such as under rubble or inside deformed vehicles. This challenge has different degrees of difficulty, depending on how hidden the victim is. For instance, in building collapses, victims that are on the top floors are easier to detect and rescue than those that are on lower floors [15], which are more isolated under a thicker layer of rubble. In traffic accidents, the type of vehicle, passive safety systems, speed, etc., play a central role in the outcome of the rescue mission.Victims often cannot collaborate, if they are incapacitated due to wounds or being trapped. In the worst case, the victims may be unconscious, making them harder to find.The lack of support infrastructure, such as pavement to ease the evacuation of victims in ambulances, cell towers that allow telecommunications for coordination [16], electricity, etc.Time is often limited [14] to find and rescue victims, which may be wounded and need medical attention. As time increases, the physical and psychological pressure on victims may cause permanent damage [17].First responders (and victims) are subject to hazards such as falling structures, flammable and/or toxic gas leaks, replications of the disaster, etc. Therefore, there is an even higher need for rescue missions to succeed in the shortest possible time.

Depending on the kind of disaster and the specific context of the victim, they can be classified into two separate groups: surface and underground victims.

Surface victims are those that end up exposed above possible debris. This is common in situations such as floods, terrorist attacks, or earthquakes. Localization needs to be performed in two dimensions and with a relatively low precision, since once the first responders are nearby, they can easily locate and access the victim. Surface victims can be located with relatively simple techniques, such as human visualization or detection with rescue dogs [3]. More recent IT-based solutions have been proposed, such as audio- [14] or image-based [18] based human detection from Unmanned Aerial Vehicles (UAVs). These techniques use the signals from the UAV’s sensors and process them in real-time with techniques such as life signs detectors using drones in disaster zones supported by deep learning [19], which, paired with the GNSS location, can help to pinpoint the victims on a map. A detailed description of such a system for flood victims was described in [20]. Another prototype has emerged for surface extraction that places an autonomous wristband on the victims for monitoring their vital signs and easing the rescue [21].

In contrast, underground victims are extremely challenging to find due to visibility and sound being blocked by rubble. In [14], they implemented an audio-processing-based human detector with a UAV even under rubble. The victims needed to be located in three dimensions, in order to better assess their situation, communicate with them, and estimate their chances of survival. Moreover, the precision requirements were higher, since rescuing underground victims often involves complex and risky procedures to liberate them. Finding Individuals for Disaster and Emergency Response (FINDER) uses radar for detecting heartbeats or breathing variations of the victims [22]. However, underground techniques are extremely complex and cover a limited area. To overcome this limitation, the DronAid system proposed in [15] uses Passive Infrared (PIR) sensors mounted on a UAV to scan the rubble, looking for victims trapped near the surface. Table 1 compares the different victim detection and localization systems described above compared with the proposed method. The crucial characteristics that we compared were: the capability of finding individuals under rubble, a fast deployment to find victims, and the precision of the victim’s location.

## 3. Overview of Location Technologies

In this section, an overview of precise technologies for positioning, which have been designed for challenging scenarios such as indoors, are described below.

### 3.1. Cellular-Based Radio

Cellular networks are currently widely used and ubiquitous, making them present in most disaster scenarios [24]. New functionality has been added generation after generation, and currently, the Fifth-Generation (5G) is being deployed around the world. The availability of radio signals, which can be measured and used for ranging, makes them an ideal candidate for detection and localization of personal devices in disaster scenarios. 5G works on the 700 MHz, 3.5 GHz, and millimeter waves of the 26 and 28 GHz bands. Higher frequencies allow high-precision ranging in direct Line of Sight (LoS) with the target, but highly suffer from attenuation, multipath, and reflections in Non-Line of Sight (NLoS). In contrast, lower frequencies are more robust to attenuation, reaching longer distances; however, multipath effects can deteriorate the precision of the ranges. In [25], in order to eliminate the need for clock synchronization, the use of different timing techniques such Round-Trip Time (RTT) was proposed for indoor localization.

5G NR with millimeter waves fulfills the specifications of Release 16 [26], which requires a localization error below ≤3 m in the horizontal and vertical planes in indoor deployments and ≤10 m in the horizontal plane and ≤3 m the vertical plane outdoors [27]. To the best of our knowledge and due to the early-stage deployment of 5G millimeter wave technology, there are no experiments with real data that measure the real accuracy of the system, despite the existing terminals that work with mmWave [28]. However, in [29], a study based on simulations concluded that the final accuracy fulfilled the requirements. Hence, in case the 5G network cannot provide coverage where the LTE network can, UEs may benefit from LTE as a backup for other location technologies [13,30]. End-users may benefit from cellular localization in cases where no high-precision technologies are present. Older cellular generations, such as 4G, usually provide a lower precision compared to 5G. For instance, LTE utilizes the Received Signal Strength Indicator (RSSI) for ranging. RSSI highly suffers from multipath and fading, which lead to high variations and an increase in the ranging error.

### 3.2. Ultra-Wide Band

UWB stands out as one of the most-promising technologies for indoor localization [31]. It is becoming a de facto standard for indoor localization, with a growing adoption in the market [7]. UWB technology utilizes very short pulses (in the order of nanoseconds), which are translated into a wide bandwidth. This enables high data transmission rates and high-precision ranging with good obstacle penetration capabilities [32] and robustness against multipath effects in NLOS conditions [33], making UWB outstanding at detection and positioning in emergency scenarios. Thus, UWB has been previously indicated as a possible human detection technology in complex environments [34] or as an impulse radar [35].

### 3.3. WiFi

WiFi is another ubiquitous wireless technology (although mainly in indoor scenarios and with small deployments) used for communications. It is made up of different versions of the IEEE802.11 protocol family, which are supported by a very wide base of consumer devices for communications. Two different approaches of WiFi are widely implemented for location services.

#### 3.3.1. WiFi Fingerprinting-Based Localization

This is one of the most commonly used technologies and algorithms used for indoor localization [36] due to the ubiquity of WiFi networks and its low cost. Fingerprinting consists of two phases: offline training phase and online operating phase. During the offline phase, a radio map divides the scenario into a lattice, and the RSSI of the visible routers at each point of the lattice is stored. In the online phase, the system estimates the UE position, comparing the RSSI information to the most similar entry of the radio map. However, this technique cannot be used in emergency cases due to the changes in the environment, making ranging–location systems crucial for locating victims. The main issue of all fingerprinting techniques in emergency scenarios is the fact that one cannot create a radio map in the offline phase. When the catastrophe comes, such as an earthquake or flooding, the scenario completely changes, and the radio map becomes useless for localization service.

#### 3.3.2. WiFi Fine Time Measurement

More recently, the IEEE 802.11mc variant added a new Fine Time Measurement (FTM) functionality. FTM includes timestamped packets and calculates distances to the User Equipment (UE) accurately with the Round-Trip Time (RTT) protocol [12]. This may be very useful for emergency cases because the number of smartphones that include support for this protocol is increasing [9]. The distance calculation to every router that supports the FTM and RTT protocol is computed in the UE for privacy preservation, even if the UE is not connected to the router.

### 3.4. Bluetooth Low-Energy

Bluetooth Low-Energy (BLE) is also another omnipresent wireless technology used in Personal Area Networks (PANs) for data transmission. Several studies have researched within the scope of detecting, localizing, and tracking people in indoor scenarios [37,38,39]. Most of these studies work in indoor spaces, applying methods such mapping and fingerprinting, which require previous knowledge of the terrain. In this work, we assumed a potential collapse of the infrastructure due to flooding or an earthquake, which makes fingerprinting techniques unfeasible for localizing victims, due to the outdated maps collected prior to the collapse.

## 4. Victim Detection

Immediately after a catastrophe, a key aspect is to detect and account for the victims trapped under the rubble, after which localization can be performed to further concentrate the efforts of the first responders. Several techniques have been proposed for this task:Visual recognition with rescue dogs: the traditional task of finding victim is usually performed either by direct observation or with the help of trained rescue dogs [3].Human body image detection: when the victims are incapacitated, image recognition may help to find individuals that are on the surface [19,21].Audio-processing-based human detection: when victims are trapped under rubble and there is no visual or imaging recognition, drones may integrate microphones to detect any distress calls and notify the rescue services [14].Vital signs detection: Passive Infrared (PIR) sensor for detecting victims that are buried close to the surface [15]. PIR reacts only to certain energy sources such as human body heat. Low-power microwave radar signals can be used to detect the heartbeat and breathing of underground victims. The limitation of such devices is the short coverage of some meters [22].Localization using radio signals coming from devices that victims carry (e.g., smartphones) or wear (e.g., smartwatches): this paper relies on this method for detecting and then locating the victims [23].

## 5. Victim Detection and Localization Method

Drones or Unmanned Aerial Vehicles (UAVs) will play a key role in the detection and localization of victims in emergency scenarios. Several research works have used drones for finding individuals under rubble with camera recognition [15,19,21]. Drones can also act as mobile Access Points (APs), which provide network connectivity both to the first responders and to the victims, who can potentially use the drones to transmit their information. User devices that integrate WiFi interfaces periodically send a control frame to have the nearby wireless access points’ information [40]. For this work, we exploited this scanning procedure from the user side to detect different devices in the area and listed by their MAC address. Once the victim is detected, more drones will approach the victim’s location by looking at the RSSI values. When a sufficient number of drones is in the vicinity of the victim’s location, the localization will be determined by using trilateration. Basic trilateration obtains the position of the target in 2D based on the intersection of the distances from at least three reference points; for 3D, at least four reference points will be needed. In this case, the ranging information must be obtained by timing measurements, which may enhance the accuracy of the victim’s position compared with the RSSI ranging performance. Figure 1 illustrates drones scanning the surface looking for victims under rubble (in red). Once a victim is detected (in yellow), the drone notifies a central coordination system, and the rest of the air fleet approaches the site to serve as additional reference points and to complement the partially damaged infrastructure. When the victim is located (in green), the system knows approximately the position of the victim and the emergency response will start.

To know the victim’s position, it is necessary to have the information of, at least, four reference points. The higher the number of drones providing location information, the more accurate the victim’s localization can be. Thus, fusing different technologies can help fulfil or increment the number of visible reference points to augment the accuracy. In [13], opportunistic fusion was proposed using several different technologies for ranging, specifically UWB, WiFi, and LTE. Opportunistic fusion takes advantage of the fact that, in most locations, several different radio technologies are visible and that modern mobile devices support these technologies. Instead of using the reference points and ranges obtained from a single technology (e.g., UWB), opportunistic fusion uses whichever reference points are visible to the device.

The major challenge for this system (and, in general, for localization with mobile devices in a catastrophe) is to collect the measurements of victims under rubble with missing elements in the network. In this paper, opportunistic fusion is proposed for taking advantage of whichever infrastructure is undamaged in a catastrophe (such as cellular network base stations or WiFi routers), complemented with portable reference points (such as drone-mounted WiFi access points with FTM capability or mobile UWB access points), to detect, estimate the distance to, and triangulate the approximate position of the victims. Opportunistic fusion greatly enhances the chances that the victim is within the range of four reference points and, therefore, can be located.

One of the main requirements of this system is to have access either to the UE measurements of the reference points or to the uplink measurements of the network. Either way, the inclusion of additional functionality tailored to accurate localization in the mobile network standards would greatly benefit the implementation of a system such as the one described in this paper. There is, in fact, a protocol that allows the UEs to report localization to the mobile network (from which the proposed system could extract that information) called NRPPa [26]. Currently, this precise localization is obtained with GNSS receivers in the UEs and is reported through NRPPa as estimated coordinates. To fuse different ranges and different reference points, including temporary ones, the following messages should be included [30] as an extension to NRPPa:■Reference point identifier;■Reference point location;■Technology type;■Timestamp;■Round-trip time.

## 6. Materials and Methods

Locating victims under rubble is a very challenging task, so even a gross estimation of where they may be is of great value to first responders. While radio-based methods for detection and localization may be of help, rubble is a very harsh environment for radio propagation. Therefore, there is a need to better understand propagation in this kind of environment. In this section, we insert a smartphone into a pile of rubble over 50 cm thick in all directions to simulate a buried victim. We deployed UWB, WiFi, FTM, and LTE to estimate the location of the UE and obtain accurate horizontal and 3D information.

Rubble was made of several metallic objects such as computers, chairs, desks, and bricks, as shown in Figure 2, in order to emulate a disaster scenario in which the victim has been caught under rubble. The UE used for the experiments was a Google Pixel 3, which supports WiFi FTM. For UWB, we attached a DWM1001 device from Qorvo to the UE using a Bluetooth serial port. The UE was equipped with an Android application that estimates the distances to all the reference points of any technology (UWB, WiFi FTM, and LTE) that are within coverage and sends the estimated ranges to a server, where the localization is computed solving the trilateration problem.

We expected the radio signals to be severely affected by attenuation and multipath, causing estimation errors in the ranges and, therefore, errors in the localization. We found that, for the purposes of better understanding the performance of WiFi and UWB under rubble, this emulated scenario could provide enough realism, as it could be used to assess the accuracy.

Figure 3 represents the map of the scenario where the UWB (green) and WiFi APs (orange) are placed and the distance to the device under rubble is given. The distance from the UWB (UWB1, UWB2, UWB3, and UWB4) and the WiFi (WiFi1, WiFi2, WiFi3) reference points is indicated next to the dashed line. To clarify the schema, vertical and horizontal views are provided for a better comprehension of the scenario.

The UWB reference points were also DWM 1001 devices, programmed as anchors, and the WiFi FTM was the Google WiFi routers. UWB and WiFi reference points were placed at 2.17, 2.43, 5.93, and 9.27 m away from the victim’s device, configured with their default parameters [41], and their heights were 1.16, 0.39, 0.86, and 0.45 m, respectively. The DWM1001’s power transmission is −14.3 dBm, and UWB anchors were centered in the 6 GHz frequency band [41].

WiFi FTM also works on the RTT protocol; hence, its ranging accuracy is also precise. Moreover, the implantation of the WiFi FTM chipset is widely implemented [9]. WiFi FTM APs are part of Google routers’ family [42]. WiFi APs were placed next to the UWBs with the same distance and heights from the victim’s device. WiFi FTM is centered in the 2.4 GHz band as the typically WiFi frequency and transmits with a power of 28.17 dBm [42].

The LTE network consisted of up to 12 femtocells (of which, 4 were visible to the UE) operated by the research team and located at different floors above the scenario, at 5, 9, and 15 m, and they were configured with a transmission power of −6.8 dBm and downlink and uplink frequencies of 2630 MHz and 2510 MHz, respectively.

The WiFi FTM and UWB ranges were obtained with the RTT protocol, which estimates the distances according the to propagation time. The measured distances were sent to a Flask server, which was run on a laptop with Windows 10 for processing. Measurements were captured during 5 min with a sampling rate of 1 s. With this distribution, the penetration capacity and the ranging-error of UWB and WiFi FTM are calculated.

## 7. Results

In this section, the results of the measurement campaign are presented. We compared the results obtained from the device under rubble with the ones with the device outside of the pile of rubble.

### 7.1. Performance of Different Technologies under Rubble

UWB and WiFi FTM are outstanding precise technologies for challenging scenarios such as indoors. Nevertheless, once the device is buried under rubble, the performance of the ranging estimation is degraded by severe multipath and attenuation. Figure 4 shows the ranging estimation error of the UWB anchors (UWB1, UWB2, UWB3, and UWB4) and WiFi FTM (WiFi1, WiFi2, WiFi3) in both cases: under rubble (blue) and outside of the rubble (red). The 80th percentile of error in both technologies increased considerably around 1 m. An important observation was that, during the whole measurement campaign under rubble, no packet was captured by the smartphone from a UWB reference point further than 9 m (as a consequence, UWB4 was left out of the measurements in Figure 4), in contrast with WiFi, which could capture the data without any problems.

### 7.2. Localization Degradation under Rubble

Figure 5 represents the Cumulative Distribution Function (CDF) of the localization error of the victim using the ranges analyzed in Section 7. The positioning error of the measurements under rubble (green dashed line) were compared with those of indoor localization (black line) [13] with the fusion of UWB and WiFi in both. As expected, it can be observed that the positioning accuracy worsened with the obstacles. The pink line represents the 80th percentile, and the error for this point was augmented by 14 m when the victim was under rubble.

## 8. Discussion

In this section, the results are examined and debated, showing the advantages of using multi-technology fusion for detecting and locating victims when a disaster occurs. While LTE and 5G NR may offer a very gross approximation of the localization, they have serious disadvantages: in a disaster, the fixed infrastructure may be partially or totally damaged; they offer a much lower precision when using RSSI measurements. In this work, the scenario emulated a realistic building collapse to compare the performance of UWB and WiFi in this kind of scenario. We found that WiFi performed better in detection and localization than UWB. However, the use of both technologies is still useful for locating victims. A combination of these high-precision ranging technologies helps to augment the number of ranges, which helps to deal with the interruptions of communication with UWB due to WiFi being stronger. Once the victim is detected, a fleet of drones can approach the place where the victim is located with meter-level precision.

In this work, we showed how the accuracy decreases when a disaster occurs and the victim is found under rubble, as shown in Figure 5. However, we demonstrated feasible drones that can easily integrate the small chipsets. From the victim’s point of view, the latest smartphone trends show that the UWB [7,8] and WiFi [9] chipsets are integrated in several smartphones, and the tendency is to increase this number. Thus, the cost associated with this integration is virtually null.

## 9. Conclusions

People usually typically have a smartphone on them, and in the case of a disaster, such as an earthquake, these devices could be very helpful for finding individuals. Drones are terrain independent and can easily approach a victim’s location. The integration of WiFi FTM and UWB with the drones could be crucial for first responder activities to detect victims under rubble. Despite the fact that WiFi presented better results with regard to the accuracy, coverage, and penetration capabilities, as shown in Figure 4, it is worth implementing both UWB and WiFi technologies to improve the inputs to the localization algorithm. Future lines of work are to implement more radio technologies such as BLE or cellular 4/5G due to their multiple advantages: to augment the coverage and probability of detecting and localizing victims, to have higher accuracy in the localization process, and to implement a new protocol that is more stable and faster between the first responder services and the drones. In addition, in the scope of this work, another future goal is to reproduce the experiment in a realistic emergency scenario in a bigger deployment with more realistic damaged infrastructure and the participation of first responders, such as in training installations. Moreover, a study of the integration of a real prototype within the rescue protocols of the first responders may be carried out.

## Figures and Tables

**Figure 1 sensors-22-08433-f001:**
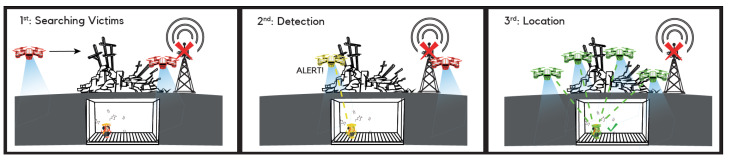
Illustration of the method.

**Figure 2 sensors-22-08433-f002:**
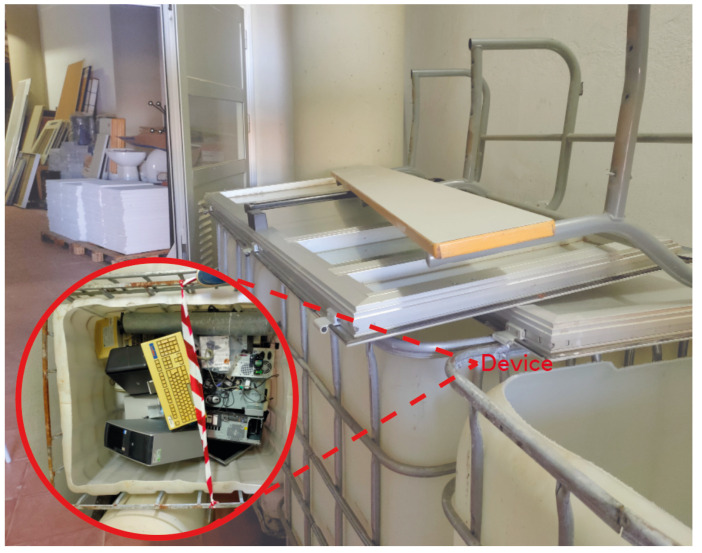
Image of the scenario.

**Figure 3 sensors-22-08433-f003:**
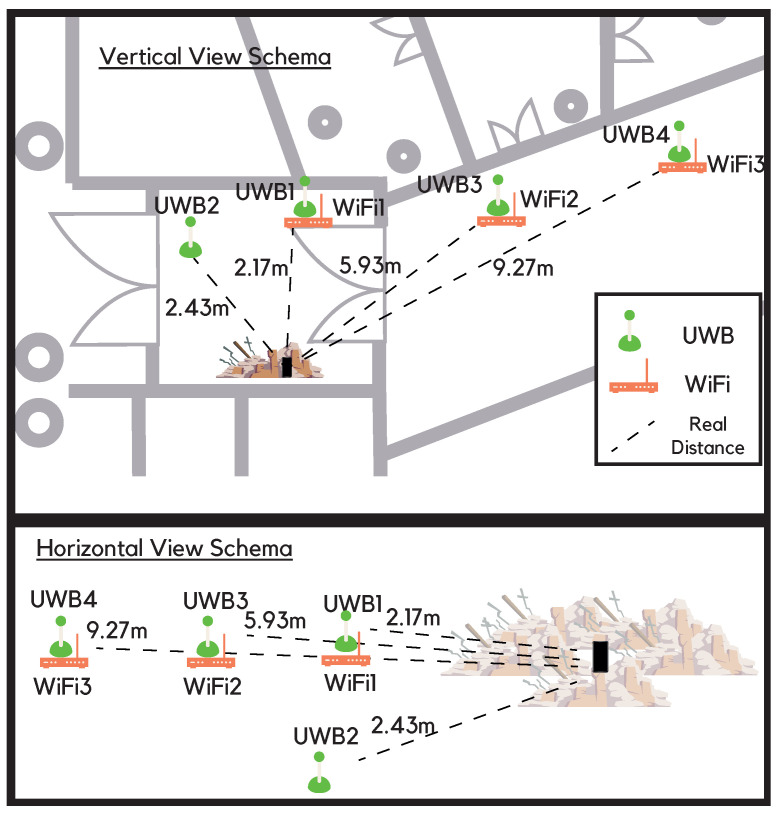
Schema of the position of the UWB and WiFi devices.

**Figure 4 sensors-22-08433-f004:**
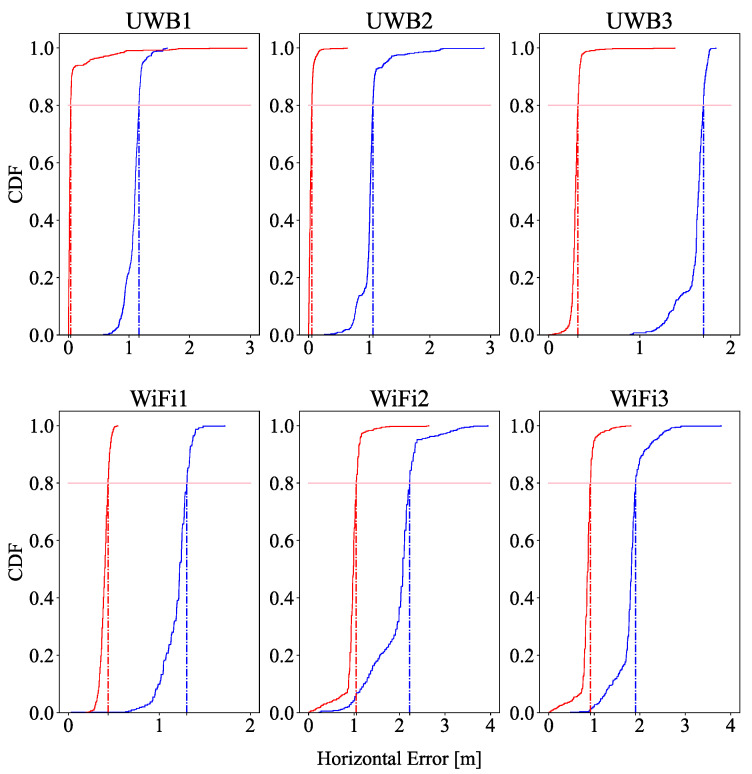
Ranging error of all the reference points with the UE under rubble (blue) and outside the rubble (red).

**Figure 5 sensors-22-08433-f005:**
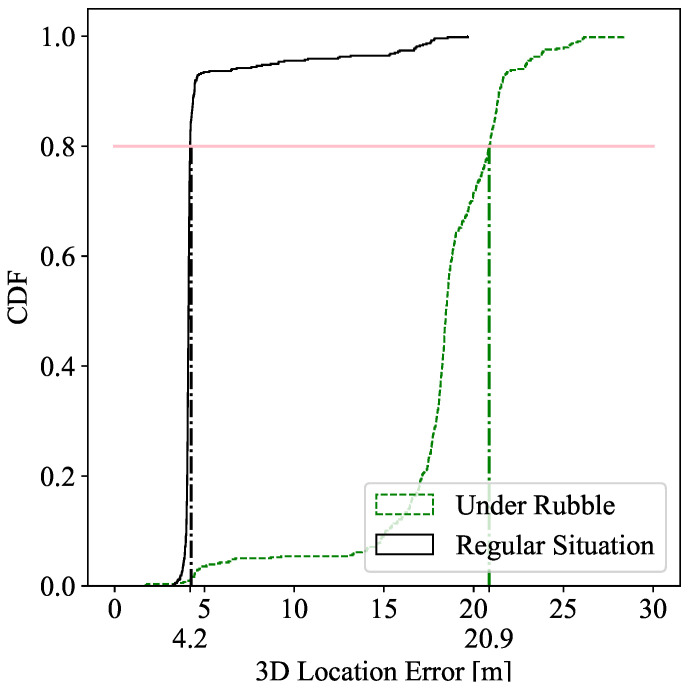
Localization performance under rubble (green) and outside of rubble (black).

**Table 1 sensors-22-08433-t001:** Overview of different methods and prototypes for the detection and localization of victims.

	Find Individuals under Rubble	Fast Finding of Individuals	Precision
Visual recognition with rescue dogs [3]	No	Yes	High
Life signs detector using a drone in disaster zones [19]	No	Yes	High
Audio-processing-based human detection in disaster sites with unmanned aerial vehicle [14]	Maybe	No	Low
DRONAID [15]	Yes	Yes	Low
Methods for autonomous wristband placement with a search-and-rescue aerial manipulator [21]	No	No	Precise
FINDER [22]	Yes	No	Precise
Victim localization using Bluetooth Low-Energy sensors [23]	Yes	Yes	Precise only at the surface
**Detection and location of victims using WiFi FTM and UWB**	**Yes**	**Yes**	**Precise**

## Data Availability

Not applicable.

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
