# Peer review of "Victim Detection and Localization in Emergencies"

_sensors, 2022, doi:10.3390/s22218433_

Round 1
Reviewer 1 Report
The authors proposed locating victims which are under rubble by fusing Fine Time Measurement (FTM) and Ultra Wide Band (UWB). Experimental results on a simulated disaster scene proved the robustness of the proposed approach. Indeed, obtaining the precise location of trapped people in a short time after disasters happen can help save their lives. The research question is meaningful and the paper can be easily followed. However, I still have some concerns about the paper.
-
There is a missing of several important approaches or techniques regarding Wi-Fi based localization, such as CSI fingerprinting [1] rather than just RSSI, and signal propagation model-based. Moreover, the overview of Wi-Fi fingerprint is insufficient, lacking many representative and latest studies, such as [2-3]
-
The author adopted one strategy to fuse the three techniques. I am wondering are there any other better fusion strategies since in [4] multiple different fusion approaches or strategies have been surveyed. This can be discussed.
-
I am wondering if the test scene is large enough to evaluate the effectiveness of the proposed approach. Normally, during emergency rescue, the searched space is much larger than the testbed.
-
My biggest concern about the study is the applicability of the proposed solution which uses UWB. As far as I know, UWB modules are rarely equipped on general smartphones. Customizing mobile devices with UWB for general users or the public also seems infeasible or costly. This should be discussed.
Other minor comments:
-
The author used ‘victim detection and location’ all through the paper. However, I think ‘localization’ should be more proper here, referring to a verb.
-
The x-axis in Figure 4 is a bit confused, such as with 0.04 under 1, and 1.16 under 1.
[1] Wang, Y., Xiu, C., Zhang, X. and Yang, D., 2018. WiFi indoor localization with CSI fingerprinting-based random forest. Sensors, 18(9), p.2869.
[2] Qin, F., Zuo, T. and Wang, X., 2021. Ccpos: Wifi fingerprint indoor positioning system based on cdae-cnn. Sensors, 21(4), p.1114.
[3] Hu, X., Shang, J., Gu, F. and Han, Q., 2015. Improving Wi-Fi indoor positioning via AP sets similarity and semi-supervised affinity propagation clustering. International Journal of Distributed Sensor Networks, 11(1), p.109642.
Author Response
The response is attached in a PDF File

Reviewer 2 Report
MDPI Sensors Journal (Manuscript ID: sensors- 1965604)
Comments to the Author
This paper investigates the WiFi Fine Time Measurement (FTM), Ultra Wide Band (UWB) and the fusion technologies for locating victims during emergencies. It is an important topic and the paper is written well, however there are several points need to be addressed to improve the quality of the manuscript.
Suggestions to improve the quality of the paper are provided below:
1) First of all, the abstract should be improved by including a few important results of the proposed approach. On top of that, the implications of this work should be included to present a broader perspective. Please enrich the abstract accordingly.
2) The authors should clearly state the novelty of the paper, which is currently missing from the manuscript. Also, the main contributions should be rephrased to more clearly highlight its novelty over the existing literature and not just a description of what was done.
3) When it comes to human detection, several indoor localisation technologies has been frequently used in the literature. One of them is definitely Bluetooth/BLE technology. The authors should compare BLE vs chosen technology (WIFI & UWB) for location detection. Some of the occupant detection studies using BLE technology have been listed below. Please review these works in the manuscript and include a short section under Section 3.
P. C. Ng, P. Spachos and K. N. Plataniotis, "COVID-19 and Your Smartphone: BLE-Based Smart Contact Tracing," in IEEE Systems Journal, vol. 15, no. 4, pp. 5367-5378, Dec. 2021, doi: 10.1109/JSYST.2021.3055675.
Tekler, Z.D., Low, R., Gunay, B., Andersen, R.K. and Blessing, L., 2020. A scalable Bluetooth Low Energy approach to identify occupancy patterns and profiles in office spaces. Building and Environment, 171, p.106681.
Baronti, Paolo, et al. "Indoor bluetooth low energy dataset for localization, tracking, occupancy, and social interaction." Sensors 18.12 (2018): 4462.
Filippoupolitis, A., Oliff, W. and Loukas, G., 2016, October. Occupancy detection for building emergency management using BLE beacons. In International Symposium on Computer and Information Sciences (pp. 233-240). Springer, Cham.
Tekler ZD, Low R, Blessing L. An alternative approach to monitor occupancy using bluetooth low energy technology in an office environment. InJournal of Physics: Conference Series 2019 Nov 1 (Vol. 1343, No. 1, p. 012116). IOP Publishing.
Huang, Ke, Ke He, and Xuecheng Du. "A hybrid method to improve the BLE-based indoor positioning in a dense bluetooth environment." Sensors 19.2 (2019): 424.
4) I strongly suggest that the authors include a brief description of the different applications of human detection in the introduction on top of victim detection to help to attract other researchers who are working on similar applications to be interested in your paper. For instance, in the building domain, many indoor localisation technologies have been proposed to enable interesting applications such as building emergency management, plug load and HVAC controls. Please include the following papers to get an idea of these applications:
Indoor localisation for building emergency management
Filippoupolitis, A., Oliff, W., & Loukas, G. (2016, December). Bluetooth low energy based occupancy detection for emergency management. In 2016 15th international conference on ubiquitous computing and communications and 2016 International Symposium on Cyberspace and Security (IUCC-CSS) (pp. 31-38). IEEE.
Indoor localisation for smart plug load control
Tekler, Zeynep Duygu, et al. "Plug-Mate: An IoT-based occupancy-driven plug load management system in smart buildings." Building and Environment 223 (2022): 109472.
Indoor localisation for smart HVAC controls
Balaji, B., Xu, J., Nwokafor, A., Gupta, R. and Agarwal, Y., 2013, November. Sentinel: occupancy based HVAC actuation using existing WiFi infrastructure within commercial buildings. In Proceedings of the 11th ACM Conference on Embedded Networked Sensor Systems (pp. 1-14).
5) The conclusion section needs to be improved by including the limitations of the current approach and future research directions of this work.
Round 2
Reviewer 1 Report
The authors have solved all my concerns and I would like to accept the paper.
Reviewer 2 Report
Thank you for addressing my comments and concerns in this manuscript. The current version is ready for publication. Great work!